# Landscape and Stand Characteristics Influence on the Bird Assemblage in *Nothofagus antarctica* Forests of Tierra del Fuego

Julieta Benitez [1], Marcelo D. Barrera [2], Yamina M. Rosas [3], Guillermo J. Martínez Pastur [1,*] and María V. Lencinas [1]

1. Laboratorio de Recursos Agroforestales, Centro Austral de Investigaciones Científicas (CADIC), Consejo Nacional de Investigaciones Científicas y Técnicas (CONICET), Houssay 200, Ushuaia 9410, Tierra del Fuego, Argentina
2. Laboratorio de Investigación de Sistemas Ecológicos y Ambientales (LISEA), Universidad Nacional de La Plata (UNLP), Calle 60 y 119, La Plata 1900, Buenos Aires, Argentina
3. Department of Geosciences and Natural Resource Management, University of Copenhagen, Rolighedsvej 23, 1958 Frederiksberg, Denmark
* Correspondence: gpastur@conicet.gov.ar; Tel.: +54-2901-422310

**Abstract:** Different variables operate simultaneously at different spatial scales, influencing community composition and species distribution. This knowledge could improve management and conservation practices in managed menaced forests. The objective of this work was to determine the influence of landscape and stand variables on the bird assemblage of the managed *Nothofagus antarctica* forest of Tierra del Fuego (Argentina). We used data from bird point counts (three or four censuses during middle summer of two consecutive years) located at 48 sites distributed at four ranches. At each site, we extracted landscape variables with Fragstat software from the forest patches, the cover classes, and the whole landscape. We also evaluated local stand characteristics, such as forest structure, ground cover, and food availability, including understory plant cover usually consumed by birds and available arthropods. Data were evaluated by detrended and canonical correspondence analyses. We found that landscape configuration (e.g., forest patch shape) and local stand variables (e.g., canopy cover) influenced bird assemblage more than landscape composition. Moreover, bird functional groups responded differently to different spatial scale variables (e.g., forest specialist species were associated with forest structure, but species that use low strata to nest and feed were associated with landscape configuration variables), demonstrating the importance of using multiple spatial scales to better understand bird species requirements. The combination of practices that promote some local characteristics (e.g., high canopy cover) and more complex landscape configurations could simultaneously favor different bird species groups and improve the effectiveness of management and conservation strategies.

**Keywords:** landscape configuration; landscape composition; bird community structure; spatial scales; Patagonia

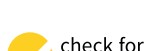



## 1. Introduction

Ecological processes that influence community composition and species distributions, such as biological interactions and habitat selection, operate simultaneously at different spatial scales [1]. At the landscape scale, composition and configuration define the heterogeneity or structure of the landscape and can generate different effects on biodiversity [2,3]. Configuration can be defined as the spatial arrangement of habitat patches within the landscape (e.g., number of forest patches and edge density), while composition can be defined as the type and proportion of the different forms of land cover (e.g., percentage of forest cover and matrix composition) [4,5]. Some authors have observed that landscape

configuration shows more variable and weaker effects on biodiversity than landscape composition in temperate forests [2], and specifically on birds. For example, Bonfim et al. demonstrate a disproportional importance of landscape composition explaining taxonomic and functional diversity of frugivorous birds in Brazilian Atlantic Forests [6]. This was explained by the habitat size hypothesis, which predicts lower relevance of habitat configuration as a predictor of diversity [7]. On the other hand, some authors have demonstrated that landscape configuration is an important predictor of bird distribution. For example, the cover and configuration of forests are important predictors of species presence, while responses are species specific because not all species vary in the same way [8–11].

In multi-scale studies, some authors observed that vegetation structure (e.g., tree density) and landscape attributes (e.g., forest area) influence local bird abundance and diversity [1,12–15]. Furthermore, it has been observed that species respond according to different spatial scales. Bhakti et al. observed that, although some species were more influenced by microhabitat characteristics (e.g., tree height and diameter), others were more related to landscape variables (e.g., total forest area, patch area, or connectivity) [16]. These differences in the response to the habitat characteristics could be related to different functional traits of the birds, such as body size, foraging strategy, or diet. For example, different authors observed that bird species that were endemic and understory insectivores, residents, and forest specialists were more affected by landscape (e.g., patch size) and stand characteristic (e.g., vegetation structure) than migratory and habitat generalists [2,13,14,16,17].

Therefore, studies that simultaneously encompass local and landscape-level aspects can better contribute to the understanding of community composition at multi-scale levels [18]. Although many studies have been carried out on the effect of landscape variables on bird communities in fragmented forests surrounded by agricultural land [19] or urbanization [20], fewer have been carried out on low intensity harvested or natural forests. In turn, few studies have considered the effects of different landscapes and localities on biodiversity, and the effects at patch scale on bird assemblages.

*Nothofagus* forests, the southernmost forests in the world, are the main ecosystem type in Tierra del Fuego province (Argentina), occupying more than 700,000 ha [21]. *Nothofagus antarctica* forests currently occupy an important area of Tierra del Fuego (25% of the total forest area) [22], which have been used for livestock grazing and firewood extraction since the first European settlement in Tierra del Fuego near 1870 [23,24]. Recently, new silvicultural proposals were developed for *N. antarctica* forests [24,25] to prevent overexploitation, overgrazing, species and habitat loss, exotic species invasion, and other threats, and to promote the preservation of these unique forests, their biodiversity, and the multiple ecosystem services provided to society [26,27]. In addition, birds at this latitude are the most abundant and diverse terrestrial vertebrates [28], and it has been demonstrated that they are affected by forest management impacts [29–32]. However, the knowledge about the main drivers in bird community assemblages at different landscape types, and the combined effect of landscape and local stand characteristics, are still poor. The aim of this study was to determine which landscape variables influence the bird assemblages (mainly passerines) in the *N. antarctica* forests of Tierra del Fuego (Argentina), and to analyze the relative importance of the landscape compared to local stand characteristics (habitat structure and food availability) for different functional groups of bird species. This information could help to delineate practices that improve the effectiveness of management and conservation strategies.

## 2. Materials and Methods

### 2.1. Study Area

This work was carried out in the central area of Tierra del Fuego province, Argentina, where monospecific *N. antarctica* forests are dominant (Figure 1). In these forests, *N. antarctica* trees reach heights of up to 18 m and age of up to 150–200 years old, growing in variable size patches (1 to 200 ha) and occupying 55 $m^2$ $ha^{-1}$ basal area on average [33]. Livestock is one of the main productive activities; therefore, overgrazing, habitat loss, and

the invasion of exotic species (e.g., *Hieracium pilosella*, *Castor canadensis*) are the main threats for forest conservation in the study area [34]. The understory of *N. antarctica* forests includes shrubs (e.g., *Berberis buxifolia*), forbs (e.g., *Galium aparine*), grasses (e.g., *Bromus unioloides*), ferns (e.g., *Blechnum penna-marina*), and mosses and liverworts. These forests are immersed in a landscape matrix that alternates with open areas such as grasslands and peatlands. Grasslands are dominated by grasses (e.g., *Festuca gracillima*, *Alopecurus magellanicus*, *Deschampsia antarctica*, *Hordeum comosum*), graminoids (e.g., *Carex* spp., *Juncus scheuchzerioides*), and forbs (e.g., *Azorella caespitose*, *Gentianella magellanica*, *Euphrasia antarctica*), while peatlands are mainly composed of *Sphagnum magellanicum* moss, associated with other species such as *Empetrum rubrum*, *Carex* spp., *Pernettya pumila*, and *Gunnera magellanica* [35]. The altitude varies between 100 and 250 m a.s.l. and terrain is wavy with gentle slopes, accentuated in foothills. At these latitudes, summer is short and cool. The growing season is approximately 5 months, and only 3 months per year are free from freezing air temperatures (below 0 °C). The mean wind speed outside forests is 8 km h$^{-1}$, with frequent windstorms in summer that reach 100 km h$^{-1}$ [36]. The annual rainfall is distributed homogeneously throughout the year, with an average of 400–600 mm yr$^{-1}$ [37].

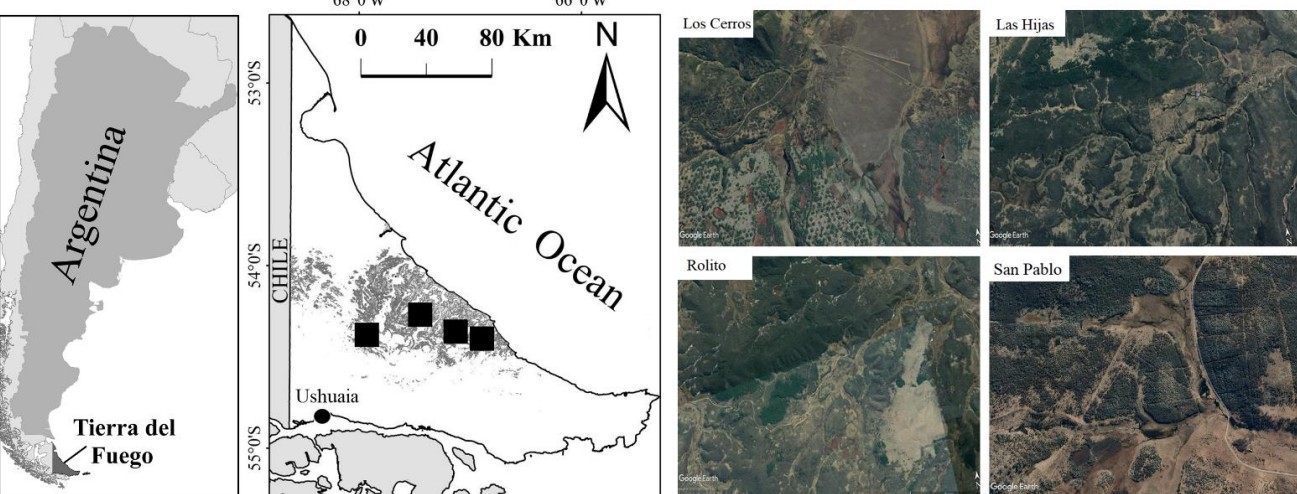

**Figure 1.** Location of study area (in black squares, from west to east: Los Cerros, Las Hijas, Rolito, and San Pablo ranches) and *Nothofagus antarctica* forests (grey) in Tierra del Fuego, Argentina. In the right panel, satellite images of the study area at each ranch.

Sampling was conducted at four locations (Figure 1) in the central area of Tierra del Fuego: Los Cerros (8900 ha; 54°20′ S, 67°52′ W), Las Hijas (10,000 ha; 54°15′ S, 67°20′ W), Rolito (16,696 ha; 54°17′ S, 67°03′ W), and San Pablo ranches (8481 ha; 54°16′ S, 66°48′ W). The *N. antarctica* forests cover 1327 ha in Los Cerros, 8100 ha in Rolito, 6300 ha in Las Hijas, and 3202 ha in San Pablo. The main productive activity in this area is livestock cattle breeding (Hereford), with an average animal density of 12 ind km$^{-2}$. Livestock use in the area dates back approximately 50 years through traditional round-year, extensive grazing in large paddocks (approximately 1000–2000 ha each) that included different ecosystem types (grasslands, peatlands, *N. antarctica* and *N. pumilio* forests), with cattle moving freely in the landscape matrix.

In some mature *N. antarctica* forest patches (1.5–13.5 ha each), forest thinning was conducted by reducing approximately 50% of the original canopy cover to increase understory forage production [32]. Similar harvesting was performed in the four ranches, eliminating mainly not healthy and suppressed-intermediate crown class individuals, and generating a homogeneous remnant and low crown cover that increase understory forage production. In this way, thinning improves the growth of the remaining trees (increasing basal area in the long term), and in the short term, it improves forage provision and the use of forests

as animal protection in winter [38]. Wood was mainly used for firewood, poles, and rods for fences.

## 2.2. Sampling Design

At each location, four habitat types dominated by *N. antarctica* forests were selected and were associated to four different canopy cover (CC) categories: (i) open forest environments (<35%) corresponding to forest edges and areas where forests advance on grasslands and peatlands, (ii) managed forests with thinning (35–65%), (iii) closed forests (65–85%) corresponding to mature or over-mature structures of primary unmanaged stands, and (iv) very closed forests (>85%) corresponding to young secondary stands with a full stocking density of trees [26]. For each canopy cover category, two replicas at Las Hijas and San Pablo and four replicas at Los Cerros and Rolito were randomly selected at each ranch, and were separated by at least 100 m from each other (N = 48).

## 2.3. Bird Sampling Methodology

Bird density was obtained with the point count method, using one observation point per replica. Each observation point was revisited 3–4 times each month during middle summer (January and February) in two consecutive years, depending on each location (2017–2020). Each visit consisted of 2 min habituation (time that birds delay in returning to normal activity, when observations are not registered) and 8 min registration (effective observation) following the methodology used by Lencinas et al. [28,30]. Bird observations were carried out during the first 4 h after dawn, when birds showed most social and feeding activities [29]. For each detected bird, as long as it was moving below the forest canopy, the species and the distance (m) to each individual from the center of the point were recorded using a TruPulse laser rangefinder (Laser Technology, Centennial, CO, USA). For bird taxonomy, we followed the South American Classification Committee [39], where species were classified by trophic level (granivores, insectivores, or omnivores) and migratory status (resident or migrant) [28,40,41].

Bird density (ind ha$^{-1}$) was obtained from censuses according to Lencinas et al. [30]. This methodology estimates the density (for each species) in one variable area, where radius is related to the habitat types, e.g., larger radius in open environments and lower in closed stands (decreasing in detectability with increasing distance). For this study, the radius for density calculation was 25 m in very closed, 32 m in closed, 35 m in thinned, and 40 m in open stands.

## 2.4. Landscape Characterization

Landscape variables were obtained using Sentinel-2 images (5 December 2016 for Los Cerros and 13 February 2017 for the other three ranches) with pixels of 10 × 10 m. For each sampling point, an individually buffer of 1-km radius was created using ArcGIS software (ESRI 2011), maintaining that point in the center, and generating a supervised classification to separate forests (*N. antarctica* and *N. pumilio*) and open areas (grassland and peatlands) (Figure 2). The landscape composition and configuration within buffers were quantified using Fragstats 4 software [42], obtaining values for three types of variables (Table 1): (1) for forest patches where the count points were found (area, perimeter, and shape), (2) for forest and open area classes (total area, number of patches, largest patch index, and connectivity), and (3) for the entire landscape (total edge and edge density). Details for units and calculations are shown in Table 1. Landscape composition variables were the total forest area and the total open areas. The landscape configuration variables were the area, perimeter, and shape of the forest patches; the number of patches; the largest patch index; and the connectivity calculated for each cover type, the total edge length, and the edge density in the landscape.

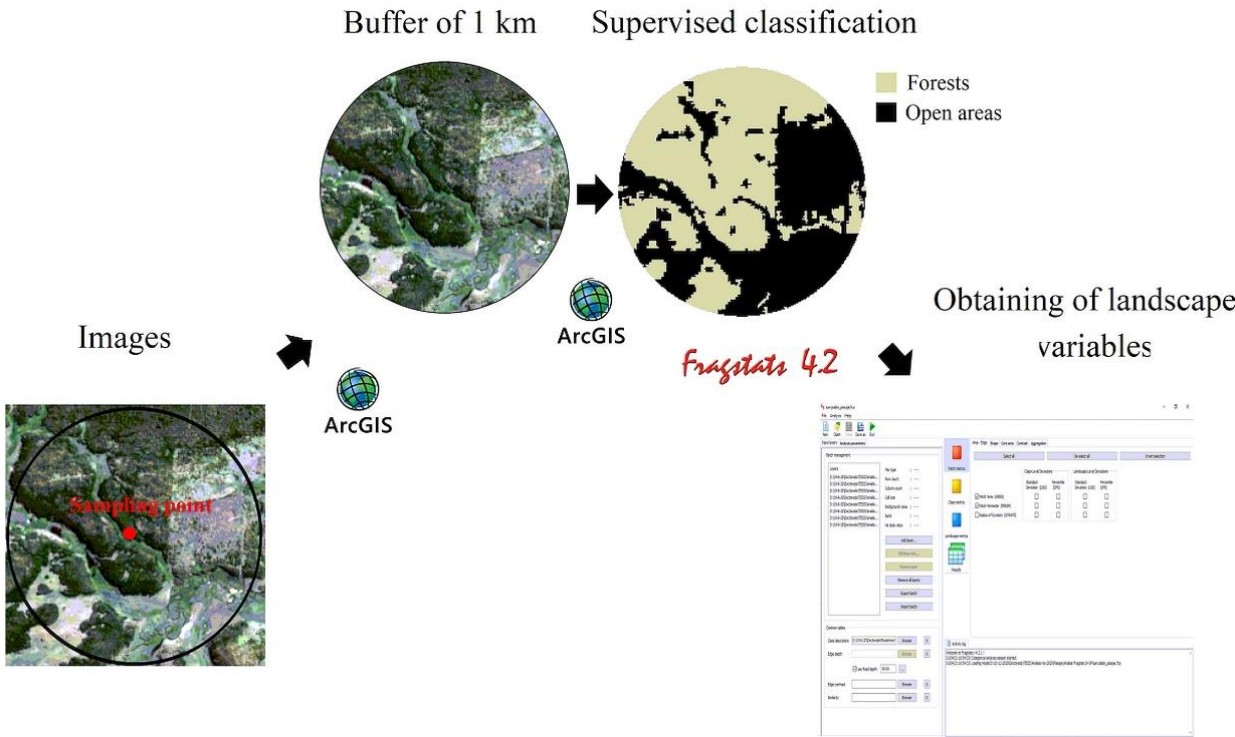

**Figure 2.** Graphical summary of the methodology to obtain landscape variables of *Nothofagus antarctica* in Tierra del Fuego, Argentina.

**Table 1.** Description of landscape variables analyzed for *Nothofagus antarctica* forests in Tierra del Fuego (Argentina).

| Type | | Variable | Acronym | Description |
|---|---|---|---|---|
| Forest patch | | Area | AREA | Total area (ha). |
| | | Perimeter | PERIM | Length of perimeter (m). |
| | | Shape | SHAPE | Perimeter of patch divided by the square root of patch area, adjusted by a constant to fit a standard square. |
| Class | Forest | Total area | FTA | Sum of areas of all forest patches, divided by 10,000 (ha). |
| | | Number of patches | FNP | Number of forest patches. |
| | | Largest patch index | FLPI | Area of largest patch of forest divided by total landscape area, multiplied by 100 (%). |
| | | Connectivity | FCONNECT | Proportion of functional joining among all forest patches (%). |
| | Open area | Total area | OTA | Sum of areas of all open area patches, divided by 10,000 (ha). |
| | | Number of patches | ONP | Number of open area patches. |
| | | Largest patch index | OLPI | Area of largest patch of forest divided by total landscape area, multiplied by 100 (%). |
| | | Connectivity | OCONNECT | Proportion of functional joining among all forest patches (%). |
| Landscape | | Total edge | TE | Sum of the lengths of all edge segments in the landscape (m). |
| | | Edge density | DE | Sum of the lengths of all edge segments in the landscape, divided by the total landscape area and multiplied by 10,000 (m/ha). |

*2.5. Local Stand Characterization*

In the sampled stands, the habitat structure was characterized, evaluating the forest structure and the ground cover. Forest structure was measured by two variable-radius plots per stand [43] with a BAF (1–7) and 50 m distance from each other. Canopy cover (CC, %) was measured with a spherical crown densiometer [44], dominant height (DH, m) with a TruPulse hypsometer-rangefinder (Laser Technology, USA), and diameter at breast height (DBH, cm) with a forest calliper, which allowed calculation of tree density (N, ind ha$^{-1}$) and basal area (BA, m$^2$ ha$^{-1}$). Ground cover (%) was measured in an area of 500 m$^2$ around each bird sampling point, using visual estimation and a modification of the "relevé" method proposed by Braun-Blanquet [45] and modified by Lencinas et al. [46]. The considered ground cover types were vascular plant species, debris (coarse and fine woody debris up to 3 cm in diameter), bryophytes (mosses, liverworts, lichens, and fungi), and bare soil without vegetation (see Lencinas et al. [46,47]). The vascular plant species were taxonomically determined following Moore [35], Correa [48], and Zuloaga et al. [49]. Based on this, the ground cover (%) of total understory plants (U), debris (D), bryophytes (B), *N. antarctica* tree saplings (TS), and bare soil (SO) were calculated for each forest stand.

To characterize potential plant food availability for birds, the cover of plant species serving as potential food for birds was estimated. Based on the information obtained from vegetation, the richness (S plants, n° species) and cover (%) of plant species potentially consumed by birds at each forest stand was calculated, and the results were analyzed as total vegetation (Veg), grasses (Grasses), and dicots (Dicots). Plant species that could be potentially consumed by birds were obtained from the bibliography [40,50–53].

To estimate the potential arthropod prey available to birds, two survey methods were employed: one sampling method using an attraction trap per forest stand, to characterize food availability at the medium-height stratum level, and a second sampling method using three pitfall traps per forest stand, to characterize food availability at the soil stratum level. The attraction traps [54] were located at 1.5 m height near the center of the stand, with a few drops of soap, water, sucrose, and vinegar as attraction agents, and these remained active for 5 days. Pitfall traps (12 cm diameter, 14 cm height) were buried at soil level, distributed approximately in a straight line (one in the geographical center of the patch and the others 5 m apart), and filled with water and soap as retention agents, remaining active for 14 days. The three pitfall traps of each stand were analyzed jointly as a single sample [55]. In the laboratory, samples were sorted by hand to identify and quantify individuals at the class and order level. We analyzed the total abundance (Ab, ind) of each trap type per forest stand, and the proportional abundance (%) of each order, which was obtained relating the order abundance to the total abundance at each trap.

*2.6. Data Analysis*

Detrended correspondence analyses (DCA) were carried out, using revisit and bird density (ind ha$^{-1}$) without down weight for rare species and with axis rescaling [56]. Then, two Canonical Correspondence Analyses (CCA, [57]) were conducted to explore the influence of landscape and local stand variables: the first one including only landscape variables, and the second one combining the two scale variables (landscape + habitat structure and metrics of food availability). A matrix with bird species density was used for each month, year, and sampling site (matrix of 13 species × 192 samples), using habitat type as a classification criterion for the different samplings. The Monte-Carlo method with 499 permutations was employed to test the significance of each axis. These multivariate analyses were performed using CANOCO 5 5.4 © Biometrics 1997–2014 [58].

## 3. Results

Twenty-one native bird species were identified, of which passerines showed the highest densities (see more details in bird species assemblages at these study sites in [32]). Therefore, only passerines (13 species, 62% of the total birds observed) were selected for the analysis. Although all of these 13 bird species have a conservation status of "least

concern" [59], many of them are endemic to the Patagonian region or are migratory and come to the area to reproduce. However, only two species are considered forest specialists (Table 2). Insectivores included the highest number of species (seven species), while the number of resident and migrant species were similar (seven species were residents and six migrants, Table 2). Eight species were common to all habitat types, but species density differed among habitat types (Table 2). *Aphrastura spinicauda* and *Elaenia albiceps* were the most abundant species in closed and very closed forests, while in thinned stands *A. spinicauda* and *Spinus barbatus* presented the highest density (Table 2). In open forests, *Tachycineta leucopyga* and *E. albiceps* were the most abundant species (Table 2).

**Table 2.** Taxonomy, acronym, trophic level, migratory status, and mean density (ind ha$^{-1}$) by habitat types for the bird species analyzed in *Nothofagus antarctica* forests of Tierra del Fuego, Argentina.

| Species | Common Name | Acronym | Trophic Level | Migratory Status | Very Closed (>85% CC) | Closed (65–85% CC) | Thinned (35–65% CC) | Open (<35% CC) |
|---|---|---|---|---|---|---|---|---|
| *Phrygilus patagonicus* | Patagonia Sierra Finch | PHPA | Omnivore | Resident | 1.7 | 1.3 | 1.3 | 3.1 |
| *Zonotrichia capensis* | Rufous-collared Sparrow | ZOCA | Omnivore | Migrant | 1.7 | 0.5 | 8.0 | 9.3 |
| *Spinus barbatus* | Black-chinned Siskin | SPBA | Granivore | Migrant | 4.2 | 9.6 | 20.4 | 6.8 |
| *Aphrastura spinicauda* * | Thorn-tailed Rayadito | APSP | Insectivore | Resident | 37.8 | 27.7 | 26.0 | 8.5 |
| *Cinclodes fuscus* | Buff-winged Cinclodes | CIFU | Insectivore | Resident | 0.0 | 0.3 | 1.1 | 1.3 |
| *Pygarrhichas albogularis* * | White-throated Treerunner | PYAL | Insectivore | Resident | 4.2 | 3.1 | 2.4 | 0.2 |
| *Tachycineta leucopyga* | Chilean Swallow | TALE | Insectivore | Migrant | 0.0 | 7.8 | 5.8 | 14.1 |
| *Curaeus curaeus* | Austral Blackbird | CUCU | Omnivore | Resident | 1.3 | 0.0 | 0.9 | 0.5 |
| *Troglodytes aedon* | House Wren | TRAE | Insectivore | Migrant | 5.1 | 3.6 | 10.6 | 7.8 |
| *Turdus falcklandii* | Austral Thrush | TUFA | Insectivore | Resident | 2.5 | 0.8 | 1.3 | 2.8 |
| *Anairetes parulus* | Tufted Tit-Tyrant | ANPA | Insectivore | Resident | 0.4 | 0.0 | 1.3 | 4.5 |
| *Elaenia albiceps* | White-crested Elaenia | ELAL | Omnivore | Migrant | 31.8 | 17.4 | 14.9 | 11.8 |
| *Xolmis pyrope* | Fire-eyed Diucon | XOPY | Omnivore | Migrant | 0.0 | 0.0 | 1.5 | 1.8 |

CC = canopy cover. * Forest specialist.

Forest patch area varied between 0.9 and 203.9 ha, perimeter between 80 and 23,040 m, and shape between 1.0 and 6.9 (Table 3). Total area, number of patches, and connectivity of forest and open areas showed similar minimum and maximum values, while the mean largest patch indexes were higher in open areas (Table 3). Local characteristics were presented by habitat types (Appendix A). Canopy cover followed the previously stated incremental gradient from open to very closed forests (26.4–95.4%), while DH was higher in thinned and closed forests (12.4 and 11.6 m, respectively). Tree density was higher in very closed forests and minimum in thinned stands, and BA followed an incremental gradient from open to very closed forests (Appendix A). Understory cover varied between 69.0% and 87.5%, with the higher values in open forests. Debris and TS were higher in thinned stands, while B was higher in open forests. Soil cover presented higher values in very closed forests (Appendix A). Richness of plants and Veg were higher in closed forest, Grasses in thinned and open forests, and Dicots in very closed forests. The abundance of arthropods captured by attraction traps, ADip, and ACole were higher in very closed, and minimum in thinned, while AHyme and ALep were higher in thinned stands (Appendix A). With respect to arthropods captured by pitfalls, PAb, PDip, and PHyme were higher in very closed forests. PAca and Plarv were higher in thinned, while PCole was higher in closed forests.

**Table 3.** Minimum, maximum, mean, and standard deviation (SD) values of landscape variables analyzed for *Nothofagus antarctica* forests in Tierra del Fuego (Argentina).

| Type | | Variable | Unit | Min | Max | Mean | SD |
|---|---|---|---|---|---|---|---|
| Forest patch | | AREA | ha | 0.9 | 203.9 | 87.1 | 60.6 |
| | | PERIM | m | 80.0 | 23,040.0 | 14,262.1 | 7497.4 |
| | | SHAPE | - | 1.0 | 6.9 | 3.9 | 1.4 |
| class | Forest | FTA | ha | 34.8 | 229.3 | 146.8 | 52.0 |
| | | FNP | - | 5.0 | 76.0 | 26.2 | 17.2 |
| | | FLPI | % | 2.7 | 64.6 | 31.1 | 16.3 |
| | | FCONNECT | % | 3.9 | 40.0 | 12.5 | 8.4 |

**Table 3.** *Cont.*

| | Type | Variable | Unit | Min | Max | Mean | SD |
|---|---|---|---|---|---|---|---|
| class | Open area | OTA | ha | 86.5 | 281.4 | 169.0 | 52.2 |
| | | ONP | - | 5.0 | 57.0 | 17.5 | 13.3 |
| | | OLPI | % | 10.4 | 72.9 | 46.6 | 16.6 |
| | | OCONNECT | % | 4.8 | 40.0 | 17.7 | 9.6 |
| Landscape | | TE | m | 18,850.0 | 43,860.0 | 29,325.6 | 5430.6 |
| | | DE | m ha$^{-1}$ | 59.6 | 138.6 | 92.9 | 17.2 |

Variable acronyms in Table 1.

In the DCA analysis (Figure 3), there is not a clear split between plots, although closed forests were located closer to very closed forest compared to thinned and open forests, and thinned were intermingled with the others habitat types. Concerning species, some migrant omnivores (*Z. capensis* and *X. pyrope*), resident (*A. parulus* and *C. fuscus*), and migrant insectivores (*T. leucopyga*) were located closer to thinned and open forests, while the two resident forest specialists (*A. spinicauda* and *Pygarrhichas albogularis*) were located closer to closed and very closed forests. Others resident omnivores (e.g., *P. patagonicus*) and a migrant (*E. albiceps* and *S. barbatus*) did not show a clear association with any habitat types.

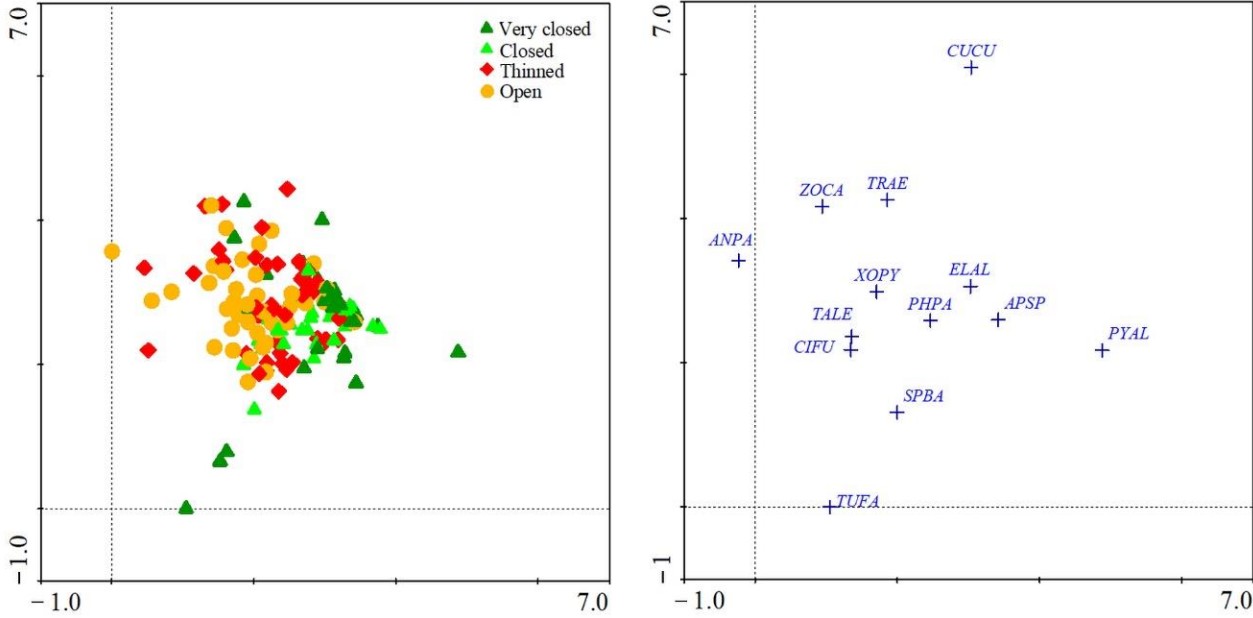

**Figure 3.** DCA ordination of *Nothofagus antarctica* forests (very closed, closed, thinned, and open) based on bird density data in Tierra del Fuego, Argentina, showing revisit in plots to the left and species to the right. Species acronyms are shown in Table 2.

When plots were ordered by CCA using landscape variables, the inclusion of explanatory variables improved from 42.3% to 84.2% for the explained total variation. Axis 1 explains 32.2% of the total variance, and Axis 2 explains 30.4% (total inertia = 3.145). Among the 13 landscape variables used in the analyses, 10 were significant for the model (Table 4), so the results are presented using only these variables as explanatory variables. The study sites of all the habitat types were intermingled, without a clear split between them (Figure 4). The two forest specialists (*P. albogularis* and *A. spinicauda*) and *A. parulus*, all resident insectivores, were associated with larger forest patches, while other migratory insectivores (e.g., *Troglodytes aedon*) were associated with higher edge density and total edge. A ground-foraging insectivore (*Cinclodes fuscus*) was associated with a higher number of forests and open area patches. On the other hand, three omnivores (*Curaeus curaeus*, *Zonotrichia capensis*, and *Xolmis pyrope*) were associated with edge density, total edge, and

number of patches of open areas, while another species of this trophic group was mostly associated with greater quantities of forest patches (*Turdus falklandii*). Only one granivore (*S. barbatus*) presented a higher abundance in landscapes with larger total open areas, while two migrant species (*E. albiceps* and *T. leucopyga*) showed little association with any landscape variable.

**Table 4.** Results of Canonical Correspondence Analysis (CCA) showing the importance of landscape variables and stand characteristics (habitat structure and food availability) in explaining variation in species composition for *Nothofagus antarctica* forests in Tierra del Fuego (Argentina), as is showed by their conditional effects. Units of variables are showed in Table 1 and Appendix A.

| Best Explanatory Variables in CCA | Acronym | Conditional Effects | | |
| --- | --- | --- | --- | --- |
| | | LambdaA | *p*-Value | F-Ratio |
| Landscape | SHAPE | 0.14 | 0.002 | 7.88 |
| | FNP | 0.07 | 0.006 | 4.09 |
| | OCONNECT | 0.06 | 0.002 | 3.48 |
| | OLPI | 0.04 | 0.004 | 2.56 |
| | PERIM | 0.05 | 0.002 | 2.94 |
| | ONP | 0.05 | 0.006 | 3.19 |
| | AREA | 0.04 | 0.028 | 2.13 |
| | ED | 0.03 | 0.024 | 2.00 |
| | OTA | 0.04 | 0.008 | 2.48 |
| | TE | 0.03 | 0.042 | 2.01 |
| Landscape and stand | CC | 0.20 | 0.002 | 11.43 |
| | SHAPE | 0.14 | 0.002 | 8.69 |
| | PAb | 0.07 | 0.002 | 3.99 |
| | AHyme | 0.06 | 0.002 | 4.24 |
| | TE | 0.07 | 0.002 | 4.12 |
| | S plant | 0.05 | 0.002 | 3.50 |
| | TS | 0.04 | 0.016 | 2.28 |
| | Veg | 0.03 | 0.012 | 2.34 |
| | AAb | 0.03 | 0.038 | 1.88 |

Landscape variable acronyms in Table 1. Stand characteristic acronyms: CC = canopy cover (%); PAb = total abundance in pitfall traps; AHyme = proportion of Hymenoptera captured by attraction traps; S plant = richness of plants consumed by birds; TS = cover of tree sapling; Veg = cover of plants consumed by birds; AAb = abundance of arthropods captured by pitfalls (see units in Appendix A).

When the landscape and the local stand variables were analyzed, the inclusion of explanatory variables improved from 42.3% to 81.7% for the explained total variation. Although the value explained by these variables was slightly lower than the previous analysis, graphically the distribution of the species and plots were better appreciated in this analysis. Axis 1 and Axis 2 explained 32.3% and 25.2%, respectively, of the total variance (total inertia = 3.145). Among the 38 variables (13 of landscape and 25 of local stand), 9 were significant for this analysis (Table 4). Plots were grouped by habitat types (Figure 4), with closed forests mixed with very closed at one side, thinned stands in the center, and open forests at the other side. Regarding bird species, it is observed that they were clearly separated into two groups. Two resident forest specialists (*P. albogularis* and *A. spinicauda*) and one migrant (*E. albiceps*) were associated with closed and very closed forests and were related to high values of canopy cover, plant species potentially consumed by birds, and the abundance of arthropods captured with pitfall traps. On the other hand, three omnivores (*Z. capensis*, *T. falcklandii*, and *X. pyrope*) and two insectivores (*A. parulus* and *C. fuscus*) were associated with open forests and higher values of tree sapling cover, as well as the proportion of Hymenoptera captured with attraction traps.

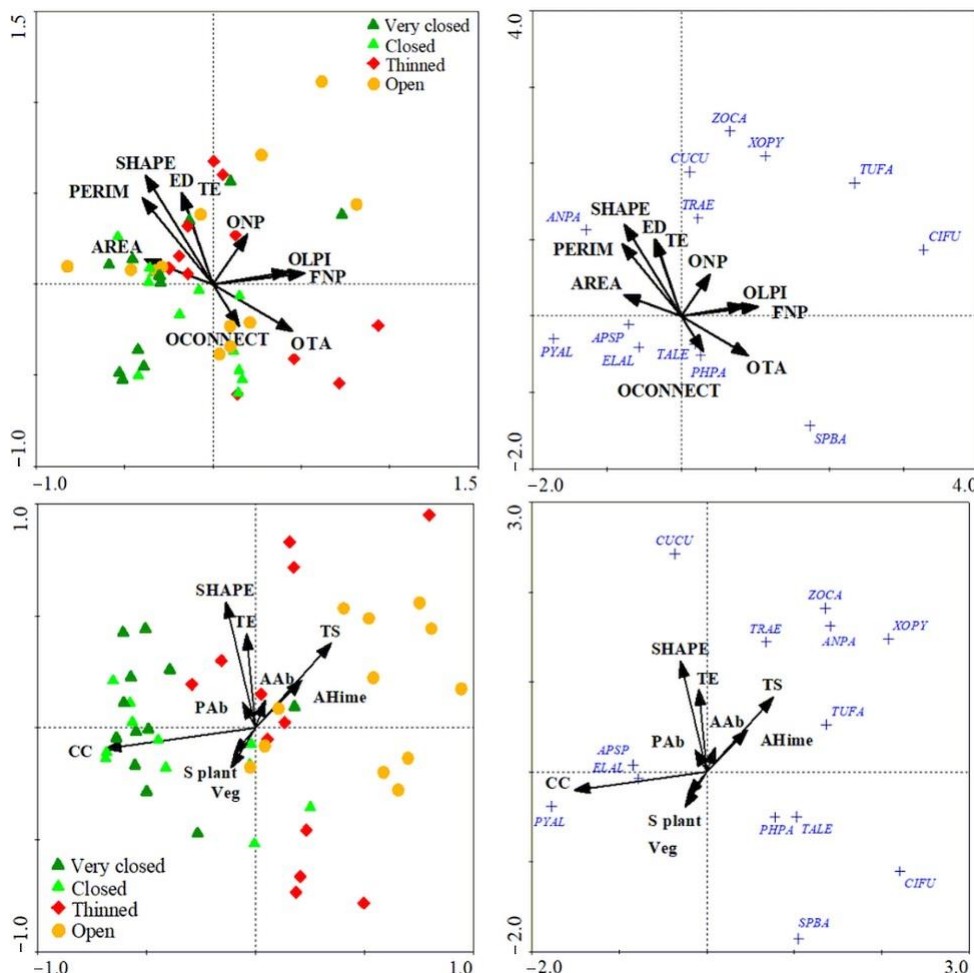

**Figure 4.** Canonical correspondence analysis (CCA) using landscape variables (superior panels) and a combination of landscape variables and stand characteristics (inferior panels) for the four analyzed *Nothofagus antarctica* forests (Very Closed, Closed, Thinned, and Open) in Tierra del Fuego, Argentina, showing plots and explanatory variables to the left, and species and explanatory variables to the right. Landscape variables: AREA = area of forest patch; PERIM = perimeter of forest patch; SHAPE = shape of forest patch; FNP = number of forest patches; OTA = surface of open areas; ONP = number of open area patches; OLPI = largest patch index of open areas; OCONNECT = connectivity of open areas; TE = total edge; DE = edge density. Stand characteristics: CC = canopy cover; TS = cover of tree sapling; S plant = richness of plants consumed by birds; Veg = cover of plant consumed by birds; AAb = abundance of arthropods captured by attraction traps; AHyme = proportion of Hymenoptera captured by attraction traps; PAb = abundance of arthropods captured by pitfalls. Species acronyms are shown in Table 2. Vectors of explanatory variables were rescaled for better representation.

## 4. Discussion

Despite the low timber aptitude of *N. antarctica* trees, these forests represent an important natural resource for firewood and poles, and they also offer a higher understory richness and biomass than other forest types in the region, which allow the development of livestock use and silvopastoral management [24], and are therefore of special interest for conservation [60,61]. Due to historically uncontrolled use, *N. antarctica* forests are significantly degraded in certain places of the Patagonia region [62–64]. In addition, the study of the bird assemblages in these forests is relevant for conservation, because in these latitudes, birds occupy many niches and key ecological roles, e.g., [65]. However, we have little knowledge about the relationships between them, and with the resources offered by *N. antarctica* forests.

Landscape configuration (area, perimeter, and shape of patches; number of patches; largest patch index) and composition (area of open areas) variables significantly influenced the bird assemblages in the studied forests, in agreement with Calamari and Zaccagnini [9] and Carrara et al. [2]. They are also in agreement with other studies, for example [7,66]; our results reject the habitat size hypothesis proposed by Fahrig [67] because it was found that not only the composition (OTA variable) but also the configuration (AREA, PERIM, SHAPE, FNP, ONP, OLPI, OCONNECT, TE, and ED) of the landscape influenced the assemblage of birds. As observed by Klingbeil and Willig [14], migratory birds (e.g., *E. albiceps* and *T. leucopyga*) showed a very weak association with landscape variables, which could be associated with the fact that these are long-distance migrants capable of adapting to many different environments [51,68]. However, a partial migratory species (*S. barbatus*) was associated with higher values of total area of open areas, while two other migrant species (*Z. capensis* and *T. aedon*) were associated with forests with more edges and a greater number of open areas patches. This could be due to these species using forests for nesting and open areas for feeding seeds and arthropods [31,50,69]. However, the species that showed the greatest influence of landscape variables were two residents and habitat generalist species: *C. fuscus* and *T. falcklandii*. These species were associated with a greater number of patches, as was also observed by Carrara et al. [2] with other habitat generalist species. On the other hand, similar results were found in other forests [2,70], where other forest specialists, such as *P. albogularis*, were associated with larger forest patches. This is an indication that this species may be sensitive to the decreasing of the forest patch area and fragmentation processes. Although *A. spinicauda*, another forest specialist species, was also influenced by the size of the forest patches, this association was weaker, probably because this specie used thinned stands and forest edges at these latitudes [32].

When we evaluate the influence of landscape variables together with the local stand characteristics, both variables influenced the bird assemblage. Landscape configuration (e.g., forest patch shape) and the habitat structure (e.g., canopy cover) and food availability (e.g., cover of plants consumed by birds) were the most associated with the composition and density of bird species. The forest specialists (*P. albogularis* and *A. spinicauda*) were associated with greater canopy cover, a characteristic that can indicate the better sites for nesting and feeding. This differs from what was observed by Smith et al. [70], who found that forest specialists are strongly associated with the total forest area in the landscape. However, this difference could be produced because the forest area around most of the sampling sites in this study is above the threshold beyond which no effects of habitat loss are observed. At the landscape level, thresholding occurs when the response of a species or group of species to habitat loss is not linear, but instead changes abruptly at a certain level of habitat loss [71]. Carrying out studies that include sampling in more areas with a lower proportion of forests could help to determine if there is a threshold from which the forest specialist species are affected, as suggested by the threshold theory. On the other hand, insectivores and omnivores that use low strata to nest and/or feed (e.g., *T. falcklandii*, *A. parulus*, and *Z. capensis*) were associated with landscape configuration variables (shape of the patch and total edge) and sapling cover in the forests. This could occur because more irregular forest patches would generate a more heterogeneous landscape that would provide greater diversity of resources for insectivores and omnivores. At the same time, high sapling cover into the forests can provides great opportunities to nest and feed for these species. As for migrant and residents, a very clear trend was not observed. It should be considered that the increasing in the length of edges due to forest fragmentation could increase the density of some residents (*A. parulus*, *C. curaeus*, and *T. falcklandii*).

These results suggest that different bird species respond to habitat characteristics at different spatial scales (local stand vs. landscape) as has also been observed by other authors [16,70]. Forest specialist species were more related to forest structure variables (canopy cover), as was observed by Muhamad et al. [72] and Menon et al. [73], and other habitat characteristics at the local stand level (understory plants and abundance of arthropofauna), as was observed be Vergara and Schlatter [74], Stratford and Stouffer [75],

and Cahill et al. [76]. Therefore, these species may be more sensitive to activities (e.g., clear cutting, thinning) that result in a reduction of canopy cover below 80% and decreasing abundance of arthropods below 1000 ind per trap. On the other hand, insectivorous or omnivorous bird species that use low strata and open areas were more associated with landscape configuration variables (e.g., more irregular forest patches, total edge) as Carrara et al. [2] observed, but also with other local variables such as the proportion of arthropods (e.g., Hymenoptera) and cover of intermediate stratum trees (saplings). Thus, these species would be more influenced by activities that reduce the lengths of edges below 30 km (e.g., due to increase or regrowth of the forests) or change the patch shape to squarer patches, but also by those that decrease sapling cover (below 3%), for example, due to high grazing intensity [77], or using ground-based extraction systems (skidders) and cable yarders [78]. The implementation or avoidance of particular practices that modify influential variables at different spatial scales can improve the effectiveness of management and conservation strategies.

## 5. Conclusions

Landscape configuration and composition variables significantly influence bird assemblages in the *Nothofagus antarctica* forests of Tierra del Fuego. When analyzed jointly with local stand variables, landscape configuration becomes more important. Canopy cover, patch shape, tree saplings, total edge, richness and cover of plants consumed by birds, abundance of arthropods, and proportion of Hymenoptera influenced the bird assemblage, demonstrating the complexity of the system. Our study highlights the importance of maintaining forests with high canopy cover to conserve forest specialist birds, and irregular parches, higher number of edges, and sapling cover to benefit insectivorous or omnivorous bird species that use low strata and open areas. The use of multiple spatial scales allows us to better understand which variables are associated with the whole assemblage and with the different bird species, as well as which modifications at local or landscape level could affect them.

**Author Contributions:** Conceptualization, J.B., M.V.L. and M.D.B.; methodology, J.B., M.V.L. and G.J.M.P.; validation, J.B.; formal analysis, J.B. and Y.M.R.; investigation, J.B. and M.V.L.; resources, J.B., M.V.L. and G.J.M.P.; data curation, J.B.; writing—original draft preparation, J.B. and M.V.L.; writing—review and editing, M.V.L., G.J.M.P. and M.D.B.; visualization, J.B.; supervision, M.V.L.; project administration, J.B. and M.V.L.; funding acquisition, J.B. and M.V.L. All authors have read and agreed to the published version of the manuscript.

**Funding:** This research was funded by ANPCyT-MINCyT (Argentina), grant number PICT-2016-1968.

**Institutional Review Board Statement:** Not applicable.

**Informed Consent Statement:** Not applicable.

**Data Availability Statement:** Not applicable.

**Acknowledgments:** This research is part of the doctoral thesis of J.B. (Faculty of Ciencias Agrarias y Forestales in the Universidad Nacional de la Plata). We would especially like to thank C. Henninger and the Luna family for allowing our entry and sampling in their ranches and I. Ramos Vértiz and A. Pellegrinuzzii for their help during sampling.

**Conflicts of Interest:** The authors declare no conflict of interest.

## Appendix A

**Table A1.** Mean and standard deviation values of stand characteristics (habitat structure and food availability) by habitat types analyzed for *Nothofagus antarctica* forests in Tierra del Fuego (Argentina).

| Stand Characteristics | | Habitat Types | | | |
|---|---|---|---|---|---|
| | | Very Closed (>85% CC) | Closed (65–85% CC) | Thinned (35–65% CC) | Open (<35% CC) |
| Habitat structure characteristics | | | | | |
| Canopy cover | CC (%) | 95.4 ± 7.3 | 82.2 ± 12.0 | 58.5 ± 12.5 | 26.4 ± 9.6 |
| Dominant height | DH (m) | 10.1 ± 2.2 | 11.6 ± 3.0 | 12.4 ± 1.6 | 10.4 ± 136.8 |
| Diameter at breast height | DBH (cm) | 18.6 ± 4.6 | 37.5 ± 10.7 | 41.8 ± 9.6 | 19.0 ± 6.3 |
| Tree density | N (ind ha$^{-1}$) | 2892.7 ± 1332.2 | 626.3 ± 533.0 | 249.8 ± 136.8 | 416.4 ± 13.1 |
| Basal area | BA (m$^2$ ha$^{-1}$) | 47.1 ± 16.6 | 34.3 ± 6.6 | 22.3 ± 6.3 | 7.3 ± 9.3 |
| Understory plants cover | U (%) | 69.0 ± 20.9 | 73.4 ± 5.6 | 77.0 ± 19.5 | 87.5 ± 12.3 |
| Debris cover | D (%) | 14.4 ± 9.5 | 14.8 ± 4.0 | 14.9 ± 9.3 | 1.4 ± 7.2 |
| Bryophytes cover | B (%) | 4.0 ± 1.3 | 7.1 ± 4.0 | 3.2 ± 3.8 | 8.4 ± 6.4 |
| Tree saplings cover | TS (%) | 0.3 ± 0.9 | 1.4 ± 2.6 | 5.0 ± 7.2 | 4.0 ± 1.4 |
| Bare soil cover | SO (%) | 12.6 ± 11.9 | 4.8 ± 2.8 | 4.9 ± 3.0 | 2.7 ± 3.8 |
| Food availability characteristics | | | | | |
| Richness of plants | S plant (n° species) | 6.3 ± 1.7 | 9.1 ± 2.1 | 8.7 ± 1.4 | 6.8 ± 2.9 |
| Plants cover | Veg (%) | 49.6 ± 19.5 | 50.2 ± 7.7 | 49.3 ± 6.4 | 41.0 ± 21.0 |
| Grasses cover | Grasses (%) | 30.7 ± 23.1 | 36.7 ± 11.9 | 36.9 ± 4.7 | 36.9 ± 19.7 |
| Dicots cover | Dicots (%) | 18.9 ± 16.4 | 13.5 ± 9.9 | 12.4 ± 9.0 | 3.6 ± 4.3 |
| Abundance of arthropods captured by attraction traps | AAb (n° ind) | 3202.1 ± 4826.5 | 1845.4 ± 2457.8 | 367.0 ± 320.0 | 3080.3 ± 3657.8 |
| Proportion of Diptera captured by attraction traps | ADip (%) | 93.3 ± 4.4 | 90.0 ± 8.8 | 70.5 ± 17.7 | 87.9 ± 9.9 |
| Proportion of Hymenoptera captured by attraction traps | AHyme (%) | 2.2 ± 2.2 | 5.0 ± 7.1 | 20.5 ± 19.6 | 4.4 ± 6.3 |
| Proportion of Lepidoptera captured by attraction traps | ALep (%) | 3.1 ± 3.3 | 4.0 ± 3.0 | 7.4 ± 9.2 | 7.2 ± 9.2 |
| Proportion of Coleoptera captured by attraction traps | ACole (%) | 1.3 ± 1.6 | 0.9 ± 1.4 | 1.2 ± 1.6 | 0.3 ± 0.3 |
| Abundance of arthropods captured by pitfalls | PAb (n° ind) | 1100.3 ± 1262.0 | 303.8 ± 258.8 | 299.5 ± 223.8 | 536.6 ± 640.5 |
| Proportion of Diptera captured by pitfalls | PDip (%) | 74.2 ± 16.0 | 56.0 ± 14.4 | 45.1 ± 23.2 | 63.2 ± 20.5 |
| Proportion of Hymenoptera captured by pitfalls | PHyme (%) | 6.7 ± 4.2 | 4.1 ± 5.7 | 2.6 ± 3.0 | 4.1 ± 3.2 |
| Proportion of Acarina captured by pitfalls | PAca (%) | 5.1 ± 6.2 | 4.9 ± 4.3 | 10.2 ± 9.1 | 6.9 ± 7.5 |
| Proportion of Coleoptera captured by pitfalls | PCole (%) | 5.0 ± 3.8 | 16.6 ± 9.1 | 14.2 ± 10.7 | 12.2 ± 8.3 |
| Proportion of larvae captured by pitfalls | PLarv (%) | 3.8 ± 3.8 | 4.9 ± 3.6 | 9.0 ± 10.2 | 3.4 ± 3.1 |

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
