# Peer review of "Landscape and Stand Characteristics Influence on the Bird Assemblage in Nothofagus antarctica Forests of Tierra del Fuego"

_land, doi:10.3390/land11081332_

Round 1
Reviewer 1 Report
General comments
The MS examines the relationships between bird communities and the composition and configuration of study plots. The authors have done extensive field research and analytical biotic data collection to model relationships between species and habitat use. The article is well structured, although it could be more detailed about the specific preferences of each species groups in the results, perhaps with some more tables or charts, in addition to the CCA ones, to lighten the discussion a bit (where similarities and differences to other studies are more easily perceived).
Minor comments
Comment 1
Series 55 – 57: It is not clear in which group of factors these species are more affected. It should be clarified
Comment 2
Row 84: I would suggest that information on the elevation and slopes of the study area should also be provided, to make the effect of geomorphology on the study area clearer
Comment 3
Row 90: In the study area, a larger map of Argentina identifying the location of the study area will also be needed
Comment 4
Section 2.4. Landscape characterization.
As landscape unit, each replica was calculated individually? It will take more explanation
Comment 5
Section 2.6. Data analysis.
I would suggest that an initial DCA analysis should be presented to examine the overall inertia based on species distribution variability alone, before constraining it, using environmental variables, to see how much variation remains unexplained
Author Response
R1
General comments
The MS examines the relationships between bird communities and the composition and configuration of study plots. The authors have done extensive field research and analytical biotic data collection to model relationships between species and habitat use. The article is well structured, although it could be more detailed about the specific preferences of each species groups in the results, perhaps with some more tables or charts, in addition to the CCA ones, to lighten the discussion a bit (where similarities and differences to other studies are more easily perceived).
A: Thanks for the suggestion, we added a DCA analysis to show specific preferences of each species groups as you suggest and we try to lighten the discussion. We do not add more tables because we believe that they would be too many.
Minor comments
Comment 1
Series 55 – 57: It is not clear in which group of factors these species are more affected. It should be clarified
A: This was clarified in the text.
Comment 2
Row 84: I would suggest that information on the elevation and slopes of the study area should also be provided, to make the effect of geomorphology on the study area clearer
A: We added this information in the text.
Comment 3
Row 90: In the study area, a larger map of Argentina identifying the location of the study area will also be needed
A: We added a map of Argentina in Figure 1.
Comment 4
Section 2.4. Landscape characterization.
As landscape unit, each replica was calculated individually? It will take more explanation
A: Yes, each replica was calculated individually. We clarified it in the text.
Comment 5
Section 2.6. Data analysis.
I would suggest that an initial DCA analysis should be presented to examine the overall inertia based on species distribution variability alone, before constraining it, using environmental variables, to see how much variation remains unexplained.
A: We added a DCA analysis as you suggest.
Reviewer 2 Report
In the introduction can be included more information why excatly that forests stands, area and birds assemblage have been chosen. Why not forest stnds much richer of birds in the warmer parts of South America?
Author Response
R2
Comments and Suggestions for Authors
In the introduction can be included more information why excatly that forests stands, area and birds assemblage have been chosen. Why not forest stnds much richer of birds in the warmer parts of South America?
A: We added more information about why that forests and bird assemblages have been chosen in the introduction.
Reviewer 3 Report
The aim of this study was to determine which landscape variables influence bird assemblages (mainly passerines) in Nothofagus antarctica forests of Tierra del Fuego (Argentina), and to analyse the relative importance of the landscape compared to local stand characteristics (habitat structure and food availability) for different bird functional groups. The methodology consisted of point counts (3-4 counts, mid-summer, 2 consecutive years per site) at 48 sites at 4 cattle ranches. In total, 13 bird species were counted. It is an interesting study with information relevant to bird conservation and forest management at Tierra del Fuego. Following are (i) some general comments, (ii) some more specific comments and (iii) an attachment with language corrections:
(i) General comments
1. It would be interesting and relevant to include more specific information about the bird species observed in the study. For example, since the point counts only took place in summer, what percentage of bird species using these forests did the 13 species found in this study represent? What is their conservation status? Is their ecology well known? Did this study contribute important information concerning any little known species?
2. Similarly to the above, it would be relevant and interesting to see additional information about the Tierra del Fuego. Is it a protected area? What is the main land use? (cattle ranches, timber? etc). What are the main conservation threats? How is it being managed?
3. In line with the above comments, the Abstract is a bit general. It can be enriched with specific information concerning the contribution of the study to bird conservation and forest management. What is the relevance/importance of this study for the conservation of particular bird species and the Tierra del Fuego.
4. Also in line with the above comments, the Discussion is also a bit general. Please include a paragraph describing the importance of the Tierra del Fuego (for conservation, livelihoods, farming, timber production (?) etc). Also, information about its importance for biodiversity and birds. And the main conservation and management issues affecting the area.
5. In the discussion, there is some repetition of ideas in paragraphs 1 and 2. Please edit accordingly.
(ii) More specific comments
1. In Table 2, include the common names of the birds too
2. The information in Lines 215-220 is a repetition of the numbers in Table 3. This is not necessary since the reader can find these numbers in the Table. Instead, summarize the most important information.
3. Line 249: 'with great edge density', do you mean higher edge density? Please clarify in the text.
4. Line 250: 'with greater number of forest and open area patches', it is better to say 'with a higher number of forests and open area patches'. Please correct in the text.
5. Line 255: 'with greater abundance', it is better to say 'with a higher abundance'. Please correct in the text.
6. Line 255: with ‘greater total open areas’ do you mean a higher number of total open areas, or total open areas that are larger in size? Please clarify in the text.
(iii) Language corrections: please see the attached document

Author Response
R3
(i) General comments
- It would be interesting and relevant to include more specific information about the bird species observed in the study. For example, since the point counts only took place in summer, what percentage of bird species using these forests did the 13 species found in this study represent? What is their conservation status? Is their ecology well known? Did this study contribute important information concerning any little known species?
A: We added more information about the bird species in results and discussion.
- Similarly to the above, it would be relevant and interesting to see additional information about the Tierra del Fuego. Is it a protected area? What is the main land use? (cattle ranches, timber? etc). What are the main conservation threats? How is it being managed?
A: Tierra del Fuego is a province of Argentina that present steppe at north and forest to the south. Because land use and conservation threats are different depending on the forest type, we added more of this information specifically about Nothofagus antarctica forest of Tierra del Fuego.
- In line with the above comments, the Abstract is a bit general. It can be enriched with specific information concerning the contribution of the study to bird conservation and forest management. What is the relevance/importance of this study for the conservation of particular bird species and the Tierra del Fuego.
A: We added some more information concerning the contribution of this study in the abstract but with this, we exceed the word limit. We hope this could not be a problem.
- Also in line with the above comments, the Discussion is also a bit general. Please include a paragraph describing the importance of the Tierra del Fuego (for conservation, livelihoods, farming, timber production (?) etc). Also, information about its importance for biodiversity and birds. And the main conservation and management issues affecting the area.
A: We added a paragraph describing the importance of N. antarctica forests of Tierra del Fuego in the introduction, and then, we return to that topic in the discussion.
- In the discussion, there is some repetition of ideas in paragraphs 1 and 2. Please edit accordingly.
A: We deleted some repetition of ideas in discussion.
(ii) More specific comments
- In Table 2, include the common names of the birds too
A: We added common names in the Table 2.
- The information in Lines 215-220 is a repetition of the numbers in Table 3. This is not necessary since the reader can find these numbers in the Table. Instead, summarize the most important information.
A: We deleted repetitions and summarized the most important information in the text.
- Line 249: 'with great edge density', do you mean higher edge density? Please clarify in the text.
A: Yes, we clarify in the text.
- Line 250: 'with greater number of forest and open area patches', it is better to say 'with a higher number of forests and open area patches'. Please correct in the text.
A: We corrected it in the text.
- Line 255: 'with greater abundance', it is better to say 'with a higher abundance'. Please correct in the text.
A: We correct it in the text.
- Line 255: with ‘greater total open areas’ do you mean a higher number of total open areas, or total open areas that are larger in size? Please clarify in the text.
A: We wanted to say “larger total open areas”, this was corrected in the text.
(iii) Language corrections: please see the attached document
A: We incorporate all the modifications except one in line 65 because we were talking about effects at patch scale (plots), not about patch size; and one in line 316 because we were talking about only one species.